# Physicochemical and Sensory Properties and Shelf Life of Block-Type Processed Cheeses Fortified with Date Seeds (*Phoenix dactylifera* L.) as a Functional Food

**DOI:** 10.3390/foods12030679

**Published:** 2023-02-03

**Authors:** Nashi K. Alqahtani, Tareq M. Alnemr, Abdullah M. Alqattan, Salah M. Aleid, Hosam M. Habib

**Affiliations:** 1Department of Food and Nutrition Sciences, College of Agricultural and Food Sciences, King Faisal University, P.O. Box 400, Al-Ahsa 31982, Saudi Arabia; 2Dairy Science and Technology Department, Faculty of Agriculture, Alexandria University, Alexandria P.O. Box 21545, Egypt; 3Research & Development Department, Almarai Company, P.O. Box 8524, Riyadh 12411, Saudi Arabia; 4Research & Innovation Hub, Alamein International University (AIU), Alameen City 5060310, Egypt

**Keywords:** date fruit seeds, block-type processed cheeses, microstructure properties, texture properties, sensory properties, shelf life, byproduct, fortified fiber, substitution, food waste

## Abstract

Processed cheese has rapidly been established as a commercial product in recent years. A new ingredient, a byproduct from date fruit seed (DFS), was obtained and tested as a fortified fiber from food industrial waste in block-type processed cheese. This is the first inclusive investigation to report such a test. Different concentrations of DFS (0%, 5%, 10%, 15%, and 20%) were added to block-type processed cheese as a partial substitution for butter. The current investigation was undertaken to estimate the impact of the partial substitution of butter by DFS and its effect on the product’s quality in terms of its shelf life and physicochemical, microstructure, color, and sensory properties. Quality was assessed over a 150-day storage period. The results indicate that adding DFS to cheese increased its nutritional value due to the addition of fiber. Additionally, the texture profile of cheese was decreased in terms of hardness, adhesion, springiness, and cohesiveness. The overall structure of cheeses became less compact and had a more open cheese network, which increased with increasing DFS% and duration of storage. Moreover, DFS exhibited the darkest color with increasing ratios of supplementary DFS and duration of storage. Based on the results found in the present investigation, it was concluded that an acceptable quality of block-type processed cheese could be achieved using DFS fiber at 5% and 10% levels of fortification.

## 1. Introduction

The valorization of byproducts and industrial food waste has become the main focus of the investigation to improve the sustainability of functional food [1,2,3,4]. Food fortification increases the amount of essential micronutrients in foods, improves their nutritional quality, and provides health benefits with little or no risk [5,6]. Fortification acts as a vehicle to add a nutrient (enhancer, restorative, or additive) that is absent or present in low amounts in the food matrix [7,8]. Fortification prevents nutrient deficiencies and associated issues to balance the overall nutrient profile and restore nutrients lost during processing, increasing the appeal for consumers looking to supplement their diets. Food fortification is also a public health strategy to increase nutrient intake in the population [9,10]. Food fortification adds many primary or secondary beneficial compounds to the food chain [11,12,13]. Enrichment can prevent or minimize the risk of micronutrient deficiencies in a given population or group. Therefore, it is a process for improving nutritional status and food intake [6,14]. The effect of fortification on health depends on certain parameters, such as the level of fortification bioavailability and the amount of fortified food consumed [15,16].

The challenge of producing good-tasting functional foods within the dairy industry rises when using food waste. Numerous authors have conveyed organoleptic issues related to the fortification of byproducts in dairy foods, mostly due to bitter, rheological, and textural properties or the salty off-flavors that are typical of date fruit seeds’ phytonutrients [17,18]. Furthermore, there is a deficiency of information from the consumer’s perspective with regard to consuming food byproducts as ingredients in other foods. The understandable concerns about food safety and quality that may arise, as well as the importance of sustainability as a driver of food choice, should be studied [4,19,20,21].

The Statistics Division of the Food and Agriculture Organization of the United Nations (FAOSTAT, 2020) [22] reported that the global production of date palm fruit (*Phoenix dactylifera* L.) is around 9.5 million tons yearly. Date palm fruit processing results in the production of massive quantities of date seeds as a waste byproduct, establishing 6.11–11.50% of date fruit weight depending on the variety, grade, and quality [23,24]. In the past, date fruit seeds were utilized as soil fertilizers and animal feed [25]. Nevertheless, they have increasingly been utilized in functional foods such as beef burgers, bread, muffins, cookies, processed cheese, coffee, ketchup, gluten-free cookies and biscuits, cooking oil, cakes, jams, and bio-oil, and have also been utilized in dietary supplementation, pharmacology, and cosmetics [26,27,28,29,30,31,32,33,34,35]. Figure 1 shows products that are fortified with DFS. Furthermore, date fruit seed (DFS) was shown to be principally rich in fiber and considerable amounts of vitamins, minerals, lipids, carotenoids, and protein [36,37,38,39]. Likewise, DFS is also considered to be a source of flavonoids and polyphenols, depending on the variety. The polyphenolic and flavonoid content of DFS is approximately 51 g/kg, which is higher when equated with other polyphenol-rich foods such as flaxseeds, grapes, and green tea [40,41,42]. Nevertheless, the polyphenolic compounds present in DFS have been shown to have anti-inflammatory and antioxidant properties that could be used to prevent and treat obesity, cancer, cardiovascular diseases, and neurodegenerative conditions [23,43,44,45].

Due to a lack of studies on the integral use of DFS for prolonging the shelf life of fortified block-type processed cheeses, the objective of this study was to evaluate the microbiological characteristics, chemical composition, and physical and functional potential of DFS. Furthermore, the production of functional block-type processed cheeses enriched with DFS was studied in order to assess the impact of these fortifications on the microstructure, shelf life, and physicochemical, texture, and sensory characteristics of DFS block-type processed cheeses. This allowed for the identification of the most ideal and functional form of DFS to be consumed in order to unlock its maximum potential with acceptable sensorial properties for consumers.

## 2. Materials and Methods

### 2.1. Materials

#### 2.1.1. Plant Material and Preparation

The DFS was used as a powdered food ingredient. Date Khalas variety fruit seeds, at the Tamar stage at the end of the harvested season, were obtained from the local date processing industry of Al-Ahasa Eastern Province, Saudi Arabia. The DFS was soaked and washed in normal water to eliminate any date flesh and then dried at 50 °C under a vacuum. The dried DFS was ground using a heavy-duty grinder and a stainless-steel hammermill (Guangzhou Mingyue, Nancun, China) grinding mill. Finally, a 0.30-mm sieve was used to obtain the DFS. A final step of drying for 1 h was repeated to remove the moisture from the powder. Finally, the DFS was vacuum-packaged and kept frozen until it was used.

#### 2.1.2. Study Materials

Standard cheddar cheese (Fat/DM: 55.8 and protein 25.1%), frozen cheddar cheese (Fat/DM: 54.2 and protein 24.2%), and milk fat butter (fat: 81.7%, SNF: 1.4) were obtained from Fonterra Inc., Auckland, New Zealand. Emulsifying salt, Joha C Special, was obtained from BK Giulini, Ladenburg, Germany. Antibacterial agent, Nisin (E234), was obtained from Danisco, Copenhagen, Denmark. Antifungal agent, potassium sorbate, was obtained from Nutrinova, Sulzbach, Germany. Plate count agar medium, Baird–Parker agar medium, sulfite–cycloserine agar medium, egg yolk polymyxin agar medium, violet red bile agar medium, nutrient agar medium, and notato dextrose agar medium were purchased from HiMedia Laboratories Pvt. Ltd., Mumbai, India. Sulfuric acid 98%, H_3_BO_3_ (boric acid; reagent-grade, ≥98%; pellets (anhydrous)), phenolphthalein reagent (3,3-Bis(4-hydroxyphenyl)-1(3*H*)-isobenzofuranone), glutaraldehyde solution (Grade I, 25% in H_2_O, specially purified for use as an electron microscopy fixative), phosphate-buffered solution, osmium tetroxide (ReagentPlus^®^, 99.8%), ethyl ether (analytical-grade), ethanol (absolute for analysis EMSURE^®^ ACS, ISO, Reag), and petroleum ether (ACS reagent) were purchased from Millipore Sigma Chemical Co. (St. Louis, MO, USA).

### 2.2. Characterization of DFS and Final Products

#### 2.2.1. Proximate Analysis Composition

The proximate analysis composition of protein, moisture, ash, fat, and fiber was determined for DFS and the final products from day 0 to 5 months (monthly).

##### Total Protein

Total protein was determined according to the method described previously [46], using Kjeldahl Semi-automized Foss model 2300 (Foss Tecator, Hoganas, Sweden).

##### Moisture

All samples were analyzed for moisture content in a moisture balance with a halogen lamp heating element (MS-70, A&D Instruments Company Ltd., Abingdon, UK.), and subjected to step heating to 130 °C; persistent weight was gained at +/−0.02% precision [47,48].

##### Ash

Ash contents were determined by carbon elimination of 1 g of sample, which was ignited and incinerated in a muffle furnace at 550 °C for 2 h. The flask was removed from heat and left to cool. Two milliliters of H_2_O_2_ were added and the flask was put back in the muffle furnace for further incineration over 1 h. The total ash was expressed as a percentage of dry weight [49].

##### Total Fat

Total fat was determined according to the method described previously [50], using the Soxhlet method (Ankom XT15 Extractor, Ankom Technology, Macedon, NY, USA; Soxtec™)

##### Total Fiber

Total fiber content was measured using a modified version of the method described in [51]. Briefly, 1 g of sample and 10 mL of distilled water were added and the mixture was maintained at 100 °C for 10 min. Then, 40 mL of absolute ethanol was added and the mixture was agitated and left in ice water for 30 min. The mixture was centrifuged (Universal 320 R, Andreas Hettich GmbH & Co. KG, Tuttlingen, Germany) at 1500× *g* for 10 min at 4 °C and the residue was added to 50 mL of 85% ethanol, mixed, and centrifuged again. This step was repeated using 50 mL of absolute ethanol. Finally, the residue was dried at 80 °C for 24 h and weighed.

#### 2.2.2. Physicochemical Analysis

The pH and acidity were assessed in the final products as previously described [33]. Briefly, the pH of 20 g of cheese in 20 mL of distilled water was measured using a pH meter (Crison Instrument, Barcelona, Spain). The visual color was measured with a Hunter colorimeter model D2s A-2 (Hunter Assoc. Lab., Inc., Reston, VA, USA), in terms of a* (redness and greenness), L* (lightness), and b* (yellowness and blueness). The instrument (45°/0° geometry, 10° observer) was calibrated with a standard white and black tile followed by the measurement of the final products at different storage times [52].

#### 2.2.3. Microbiological Analysis

Microbiological counts of *Escherichia coli*, *Staphylococcus aureus*, *Clostridium perfringens*, *Bacillus cereus*, yeast and mold, and coliform in the DFS (before cooking) and block-type processed cheese samples were determined monthly, from 0 to 5 months. For this determination, 10 g of each sample of block-type processed cheese was mixed with 90 mL of sterilized Ringer solution and the diluted samples were consistently spread into a sterile Petri dish containing each culture medium using a spatula. Potato dextrose agar medium was used for yeast counts and the subsequent incubation of molds for 5 days in aerobic conditions at 25 ± 2 °C. Furthermore, the presence of *E. coli* was detected using violet red bile agar medium at 37 °C for 24 h. Violet red bile agar medium was also used to distinguish coliforms, incubated at 37 °C for 24 h. Baird–Parker agar was used to distinguish *Staphylococcus aureus*, incubated at 37 °C for 48 h. Sulfite–cycloserine agar, incubated anaerobically at 37 °C for 18–22 h, was used to distinguish *Clostridium perfringens*, and egg yolk polymyxin agar was used to distinguish *Bacillus cereus*, incubated at 30 °C for 48 h [33]. The results were counted directly as colony-forming units (CFU/g) [53].

### 2.3. Cheese-Making Procedure

The ingredients used for the manufacture of fortified block-type processed cheeses were prepared according to the formulations presented in Table 1. Totals of 5%, 10%, 15%, and 20% (*w*/*w*) of the cheese mixture butter were replaced with DFS.

The procedure for the production of fortified block-type processed cheeses is presented in the flowchart in Figure 2.

### 2.4. Texture Analysis

The texture was measured using the Brookfield Texture Analyzer (Model CT3 4500, AMETEK Brookfield, Middleboro, MA, USA). Pre-test speed, test speed, trigger type, and deformation were 1.5 mm/s, 2 mm/s, 2.5 g, and 5 mm, respectively. The following parameters were assessed by texture profile analysis (TPA), as reported previously by [33]. Hardness was selected as the force that is essential to attain a given deformation; fracture ability is the force at which the material fractures; springiness or elasticity is defined as the rate at which a deformed material returns to its un-deformed state after the deforming force is removed; and cohesiveness is defined as the quantity simulating the strength of the internal bond assembly in the body of the cheese product. The cheese samples were analyzed at a temperature of 25 ± 1 °C, with a minimum of five replicates.

### 2.5. Microstructural Analysis

To investigate the microstructural properties of the fortified block-type processed cheese samples, scanning electron microscopy (SEM) (JEOL—Japan Electron Optics Laboratory—JSM-6380 LA, JEOL USA, Inc., Peabody, MA, USA) was performed at a voltage of 30.0 kV, as previously reported by [55], with a few modifications. Briefly, to stabilize the samples, pieces of block-type processed cheeses (4 × 4 × 4 mm) were cut from the center portions of the cheese blocks and placed in a 4% (*v*/*v*) solution of glutaraldehyde in 0.1 M phosphate buffer, pH 7.2, at 4 °C for 4 h. The cheese samples were washed four times in phosphate buffer for 15 min, then stabilized by osmium tetroxide 2% as a secondary stabilizer for 4 h. Then, the samples were washed three times in cacodylate buffer, followed by dehydration using a series of ethanol solutions (from 20% to 100%). After dehydration, the samples were dried using CO_2_ at the critical point and then directly mounted on a copper stub and coated with gold up to a thickness of 400 Å in a sputter-coating unit. The resolution was 3.0 nm (30 kV, WD 8 mm, SEI), the accelerating voltage was from 0.5 to 30 kV (53 steps), and the magnification was ×5 to 300,000 (149 steps). Samples from each treatment were taken both initially and after 5 months for analysis.

### 2.6. Sensory Analysis

The sensory evaluation of fortified cheese samples was achieved at the beginning of storage time and every month for 5 months. The sensory properties of cheese samples were assessed using a five-point hedonic scale (1: dislike extremely, 5: like very much). This scale consisted of the test parameters of appearance, firmness, stickiness, breakdown, gumminess, smoothness, chewiness, flavor, and overall acceptance. The sensory attributes were measured by 20 experienced staff members of the Food and Nutrition Science Department, King Faisal University, who had previously participated in the hedonic sensory assessment test. The panelist selection criteria were as follows: (1) between 25 and 60 years of age, (2) not allergic to dairy products, (3) non-smoker, and (4) available to participate in the sensory analysis during testing time [56].

### 2.7. Statistical Analysis

All analyses were performed in triplicate (n = 3). Statistical analyses were performed using SPSS for Windows (version 25; SPSS Inc., Chicago, IL, USA). The differences in mean values among sample varieties were determined (*p* < 0.05) using a one-way analysis of variance (ANOVA). Means separation was performed using Tukey multiple range tests. Values for the mean ± standard deviation are also presented.

## 3. Results and Discussion

### 3.1. Characterization of DFS

Total fiber, protein, and fat compounds were the major components of DFS. Table 2 illustrates the chemical composition of DFS. The moisture, TS (total solids), protein, fat, fiber, and ash were 9.39 ± 0.10, 90.61 ± 3.01, 4.86 ± 0.58, 18.44 ± 1.72, 65.95 ± 0.70, and 1.36 ± 0.01, respectively. From the chemical composition of DFS, it can be concluded that the protein and fiber content influence the functional properties and agreeable texture of the fortified cheese samples during storage, and can act as good emulsifying agents [26,57]. The results in terms of the chemical composition were in the same range and similar to results that were reported previously [26,32,33,37,58].

### 3.2. Characterization of Cheeses Fortified with DFS during Storage

The average chemical composition values, moisture, pH, acidity, ash, fat, and protein of the five fortified block-type processed cheese samples are shown in Figure 3a–f. Figure 3a shows the values of moisture content; it is obvious that the values decreased after fortification with DFS. The values were 49.63 ± 0.39% for the control treatment, while, for treatments with 5%, 10%, 15%, and 20% DFS, the values were 48.34 ± 0.17, 47.99 ± 1.26, 47.66 ± 1.25, and 48.40 ± 0.5%, respectively. Moisture values were decreased (*p ≤* 0.05) compared with the control treatment. Cheese fortified with 10% and 15% DFS as a partial substitution for butter had 3.3 and 3.97% less moisture than the control treatment, respectively, whereas cheese fortified with 5% and 20% DFS possessed 2.6 and 2.48% less moisture than the control treatment, respectively.

During storage, the cheese samples showed gradual decreases (*p ≤* 0.05) in their moisture content as the storage period progressed; however, in the first month, the decreases were not significant (*p* ≥0.05). The decrease in moisture values in the control sample after five months of storage was 6.61%, while, for 5%, 10%, 15%, and 20% DFS (*w*/*w*), these values were 4.06, 2.19, 1.15, and 3.47%, respectively, after five months of storage. The change in the moisture contents can be explained due to the increase in the amount of fiber and polyphenols with increasing DFS. Dönmez et al. [59] reported that fiber and polyphenols could interact with proteins, resulting in the formation of polyphenol–protein complexes, which decreases the release of moisture contents in a concentration-dependent manner. On the other hand, the excess polyphenol concentrations increased the release of moisture.

Likewise, Figure 3b shows that the effect of the incorporation of DFS on the pH values was not significant (*p* ≥ 0.05), although a slight decrease was shown. The values were 5.17 ± 0.02 for the control treatment, while, for fortified treatment with 5%, 10%, 15%, and 20% DFS as a partial substitution for butter, these values were 5.17 ± 0.02, 5.16 ± 0.01, 5.18 ± 0.03, and 5.16 ± 0.00, respectively. During storage, the cheese samples showed slight but significant gradual decreases (*p ≤* 0.05) in their pH values due to the hydrolysis of emulsifying salts and their interaction with proteins [60]. Figure 3c shows the observations of the acidity values.

Furthermore, Figure 3d shows the effect of fortification with DFS as a partial substitution for butter on the ash values in cheese samples, which were slightly increased, although not significantly (*p* ≥ 0.05). The values were 4.23 ± 0.04 for the control treatment, while, for fortified treatments with 5%, 10%, 15%, and 20% DFS (*w*/*w*), these values were 4.36 ± 0.11, 4.26 ± 0.07, 4.47 ± 0.13, and 4.33 ± 0.05, respectively. During storage, the cheese samples showed a slight but significant gradual increase (*p ≤* 0.05) in their ash values due to the slight decrease in the moisture content.

The fat% is represented in Figure 3e, showing that the fat values in the cheese samples were slightly decreased, although not significantly (*p ≥* 0.05) for 5%, 10%, and 15% fortification with DFS, while this change was significant for 20% DFS.

The fat value of the control cheese sample was 28.00 ± 0.16%, while, for 5%, 10%, 15%, and 20% DFS, these values were 27.63 ± 1.0%, 27.33 ± 0.57%, 27.0 ± 1.00%, and 26.16 ± 0.28%, respectively. During storage, the cheese samples showed a slight but not-significant gradual increase (*p* ≥ 0.05) in their fat values, except for 20% DFS, for which this change was significant. The increase in fat values was due to the slight decrease in the moisture content.

Likewise, Figure 3f shows that the protein values in the cheese samples were slightly decreased, although not significantly (*p ≥* 0.05). The control value was 16.62 ± 0.28%, while, for 5%, 10%, 15%, and 20% DFS, these values were 16.25 ± 0.31%, 16.53 ± 0.21%, 16.53 ± 0.10%, and 16.10 ± 0.34%, respectively. During storage, the cheese samples showed a slight but not significant gradual increase (*p ≥* 0.05) in their fat values. Our data exhibit the same trend as that found in previous studies, which used rice bran, oat, inulin, and bulger for the fortification of processed cheese [60,61,62,63].

### 3.3. Microbiological Quality

Table 2 presents the microbiological quality of DFS, with values of less than 12 cfu/g for yeast and mold; furthermore, *Escherichia coli*, *Staphylococcus aureus*, *Clostridium perfringens*, and *Bacillus cereus* were not detected in DFS.

The effect of the fortification of cheese with DFS as a partial substitution for butter on the microbiological quality was evaluated during the five-month storage period. The results reveal that no pathogenic bacteria could be detected during the storage of any of the cheese treatments included in this investigation. The absence of these pathogenic bacteria may be credited to the efficiency of the heat treatment (95 °C/2 min) applied during the processing of cheese, the good hygienic practices followed during the handling and processing of cheese, or the presence of the preservatives potassium sorbate and Nisin. Our data are in agreement with previous data [63].

### 3.4. Texture Properties

The textural properties of block-type processed cheeses are provided in Figure 4a–e. Hardness is a measure of the quantity of force that is essential to compress the cheese sample and relates to the strength of the cheese matrix. By increasing DFS levels, the hardness was decreased, although not significantly (*p ≥* 0.05) [62]. The control sample had hardness values of 768 ± 18.9 and 665.8 ± 9.4, respectively, while, for 5%, 10%, 15%, and 20% DFS, these values were 717.5 ± 32.9 and 646 ± 23.8, 732.60 ± 7.5 and 639.16 ± 6.2, 636.50 ± 21.9 and 574.30 ± 20.5, and 657.60 ± 11.6 and 604.16 ± 11.2, respectively. Cheese samples (Figure 4a,b) showed a significant (*p ≤* 0.05) decrease after one month of storage, gradually showed a non-significant (*p ≥* 0.05) increase after three months of storage, showed a significant (*p ≤* 0.05) increase after four months of storage, and then dropped again significantly (*p ≤* 0.05), compared with the fresh value. The decrease in hardness may be due to the DFS, considering its considerable water- and oil-holding capacity, and also the hydration property of DFS, which may help to provide good texture and reduce syneresis and dehydration during storage [26]. Furthermore, the DFS protein can act as an emulsifier [57], which can influence the hardness of cheese.

Figure 4c presents the adhesion, which also non-significantly (*p ≥* 0.05) decreased with hardness. The control sample had an adhesion value of 0.2 ± 0.01 g, while, for 5%, 10%, 15%, and 20% DFS, these values were 0.18 ± 0.02, 0.18 ± 0.02, 0.15 ± 0.04, and 0.18 ± 0.01, respectively. During storage, control cheese samples showed a significant (*p ≤* 0.05) decrease after four months, while 5% DFS treatment showed a significant (*p ≤* 0.05) decrease in months 3 and 5. Treatment with 10% DFS showed a significant (*p ≤* 0.05) decrease only in month 4, and, with 15% DFS, showed a non-significant (*p ≥* 0.05) decrease; furthermore, 20% DFS showed a significant (*p ≤* 0.05) decrease only in months 2 and 3. Adhesion can be influenced by the emulsifying ingredient; nevertheless, DFS is considered a good emulsifying ingredient. Polysaccharides such as glucans, xylans, mannans, and cellulose, which are associated with DFS proteins, enhance their functional properties, improving the capabilities of DFS as an emulsifier [61,62].

Likewise, regarding springiness and cohesiveness, Figure 4d,e show a non-significant (*p ≥* 0.05) decrease in these values between the control sample and samples fortified with DFS. During storage, control cheese samples showed a significant (*p ≤* 0.05) decrease in springiness after one month and in cohesiveness in months 3 and 4, while cheese treated with 5%, 10%, 15%, and 20% DFS showed a significant (*p ≤* 0.05) decrease in springiness in month 1. Regarding cohesiveness, the same decreasing trend could be found, except for cheese treated with 15% DFS, which showed a significant (*p ≤* 0.05) decrease in months 2, 3, and 5, while cheese treated with 20% DFS showed a non-significant (*p ≥* 0.05) decrease.

The textural properties of processed cheese are mostly affected by its chemical composition: fat, moisture content, pH, degree of proteolysis in the raw cheese, type of cheese, and the amount of emulsifier used. Nevertheless, processing conditions such as processing temperature, stirring speed, cooling temperature, cooling rate, storage time, storage temperature, and processing time also had an effect [63]. To the best of our knowledge, there is no existing research focusing on the storage properties of the texture parameters of block-type processed cheeses with the addition of fiber from DFS as a partial substitution for butter.

### 3.5. Microstructure Properties

The microstructure properties of block-type processed cheese fortified with 5%, 10%, 15%, and 20% DFS were examined in terms of their internal microstructure by scanning electron microscopy (magnitude 500×; at 20 kV), when fresh and after five months of storage, as illustrated in Figure 5. For fresh cheese, the control cheese had a compact structure and was homogenously uniform compared with the cheeses fortified with 5%, 10%, 15%, and 20% DFS, while the fortified cheeses showed a less compact and inhomogeneous structure which increased with an increase in the % of DFS as a partial substitution for butter. Nevertheless, a filamentous-like structure could be observed through the cheese network as one of the structural characteristics of cheeses fortified with DFS. This structure appeared to have low electronic density, which might be attributed to the fiber content of DFS-fortified cheeses. After five months of storage, the overall structure of the cheeses became more open with a less compact cheese network, which increased with increasing DFS%. Furthermore, the surface of the protein matrix of fortified cheeses by DFS appeared to be coarse and the matrix itself was less compact and denser, which increased with an increase in the DFS%. These changes could probably be attributed to the protein-binding ability and emulsification capacity of added DFS. This description of the microstructure is in agreement with that previously reported for block-type processed cheese [63].

The microstructure properties of processed cheese are affected the most by the chemical composition: fat, moisture content, pH, degree of proteolysis in the raw cheese, type of cheese, and the amount of emulsifier used. Nevertheless, processing conditions such as processing temperature, stirring speed, cooling temperature, cooling rate, storage temperature, and processing time also had an effect [63].

### 3.6. Colorimetric Measurements

Color is one of the most significant visual features in dairy products, and color differences affect the storage, shelf life, and color deterioration of processed cheeses. All differences in the L*, a*, and b* color values of the samples of fortified cheese and the controls are taken into account in Table 3. DFS fortification led to significant changes in the color of the cheese; it can be seen that the lightness (L*) values of the cheese were significantly decreased in all cheese batches (*p ≤* 0.05) compared with the control. The mean L* value for the control sample was 80.96 ± 0.66, while those at 5%, 10%, 15%, and 20% DFS were 72.90 ± 0.85, 68.96 ± 2.54, 65.86 ± 4.73, and 64.45 ± 3.01, respectively. During storage, cheese samples showed a non-significant (*p ≥* 0.05) gradual decrease, except after four months of storage, which showed a significant (*p ≤* 0.05) increase.

Fresh processed cheese samples fortified with DFS exhibited the darkest color, whereas, with an increasing ratio of fortification of DFS (Figure 6), the whiteness values of the processed cheese samples decreased. During the storage period, the processed cheese became darker. This could be due to the Maillard browning reaction, which occurs during storage between the lactose and proteins in cheese [62].

On the other hand, it can be seen that the redness (a*) value of the cheeses was significantly increased in all cheese batches (*p ≤* 0.05) with increasing DFS%, compared with the control. Storage showed a significant (*p ≤* 0.05) gradual decrease, except after four- and five-month storage, which showed a significant (*p ≤* 0.05) increase.

Nevertheless, the yellowness (b*) values of cheese were significantly decreased in all cheese batches (*p ≤* 0.05) with increasing DFS% as a partial substitution for butter, compared with the control. Storage showed a significant (*p ≤* 0.05) gradual increase, except after five-month storage, which showed a significant (*p ≤* 0.05) decrease.

### 3.7. Sensorial Attributes

Since this newly-developed block-type processed cheese based on DFS fiber is destined to be consumed by humans, the acceptance of sensory attributes is an important aspect. In general, higher concentrations of DFS% and extended storage had adverse effects on the sensorial attributes of cheeses. Figure 7a–h and Figure 8 show that the appearance, firmness, stickiness, breakdown, gumminess, smoothness, chewiness, flavor, and overall acceptance gradually decreased, although not significantly (*p ≥* 0.05), with increased DFS, compared with the control. However, at 15 and 20% DFS, the decrease was significant (*p ≤* 0.05), and the breakdown in particular was decreased significantly (*p ≤* 0.05) at 20% DFS. During the storage period, there were increases and decreases in the values; however, these were not significant, except for those at 15 and 20% DFS after four and five months. The data are summarized in Figure 8 as overall acceptance.

## 4. Conclusions

In this study, we present evidence that DFS is a rich source of fiber and that it can be useful in block-type processed cheese as a partial substitution for butter. Therefore, the primary goal of the current investigation conducted on DFS was to explore its effect on the product’s quality in terms of its shelf life and physicochemical, microstructure, color, and sensory properties. Quality was assessed over a 150-day storage period. The overall structure of cheeses became more open with a less compact cheese network, which increased with increasing DFS% as a partial substitution for butter and the duration of storage. It can be concluded that an acceptable quality of block-type processed cheese could be obtained using DFS fiber at the 5% and 10% levels of fortification as a partial substitution for butter. These conclusions should support the use of DFS in functional foods and nutraceutical products. Further studies are required to assess the techno-functional properties of the final products, such as water and oil binding capacity, foaming, and emulsifying activities, in order to confirm the quality of these novel foods. Furthermore, the biological activity should be investigated, inspiring both clinical and in vivo trials of DFS.

## Figures and Tables

**Figure 1 foods-12-00679-f001:**
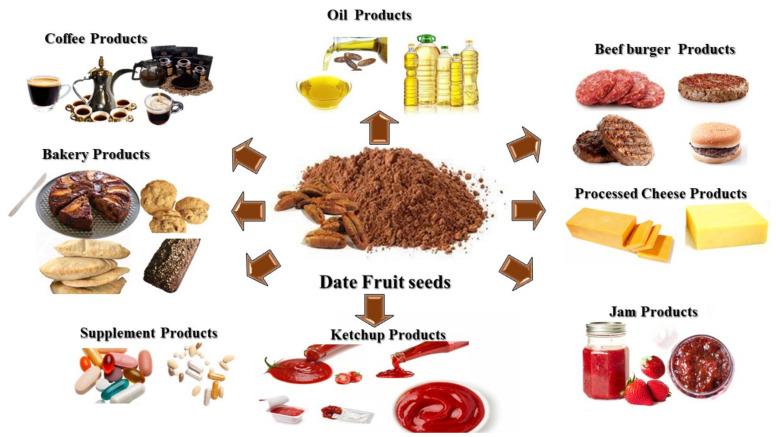
Products fortified with DFS.

**Figure 2 foods-12-00679-f002:**
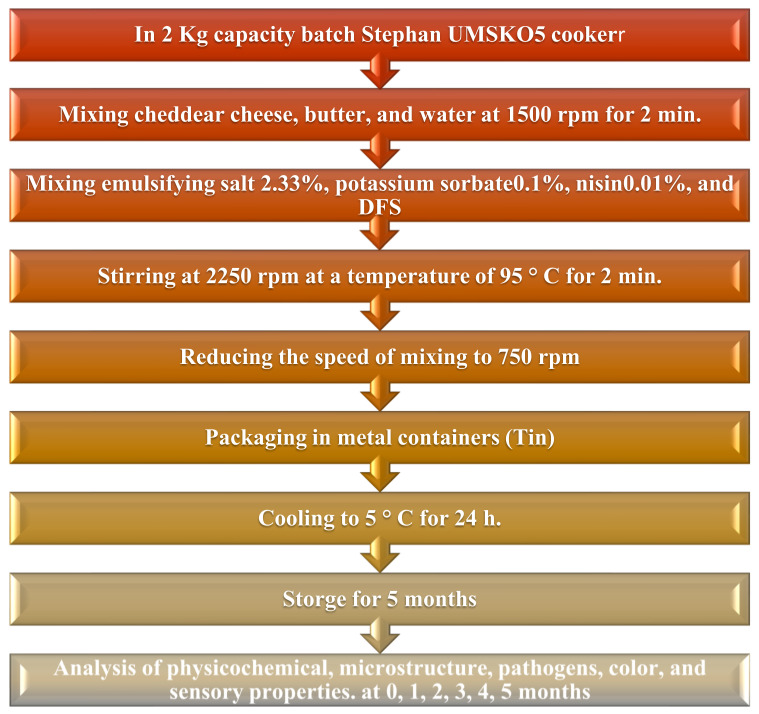
Flowchart of the effect of each ingredient in the manufacturing process of block-type processed cheeses [54].

**Figure 3 foods-12-00679-f003:**
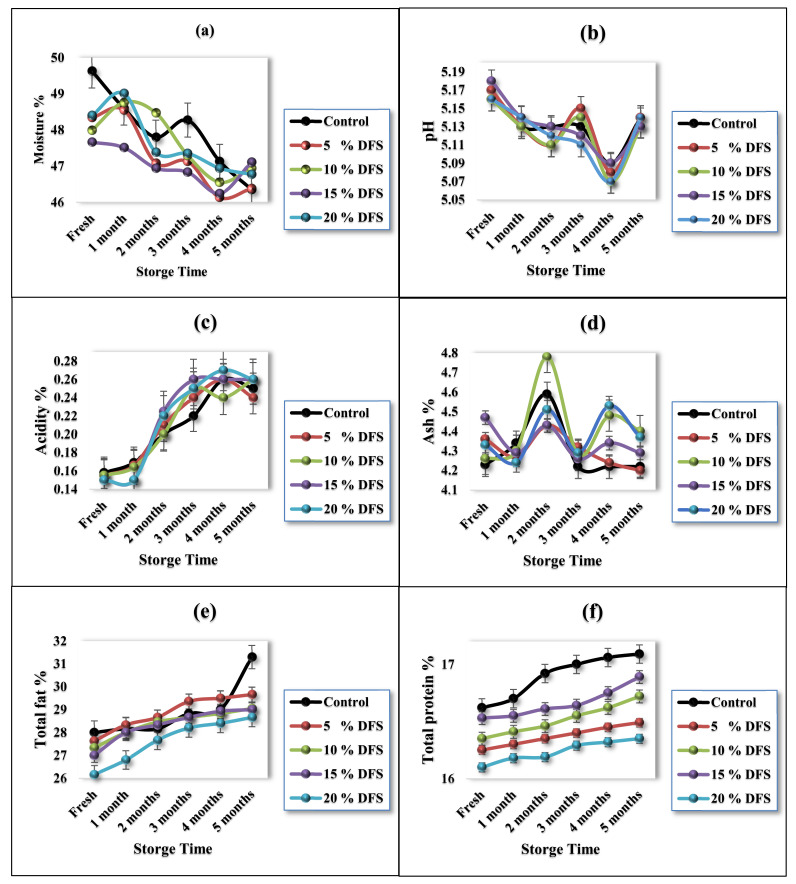
(**a**–**f**) Characterization of cheeses fortified with 5%, 10%, 15%, and 20% DFS as a partial substitution for butter in moisture, pH, acidity, ash, total fat, and total protein % during storage.

**Figure 4 foods-12-00679-f004:**
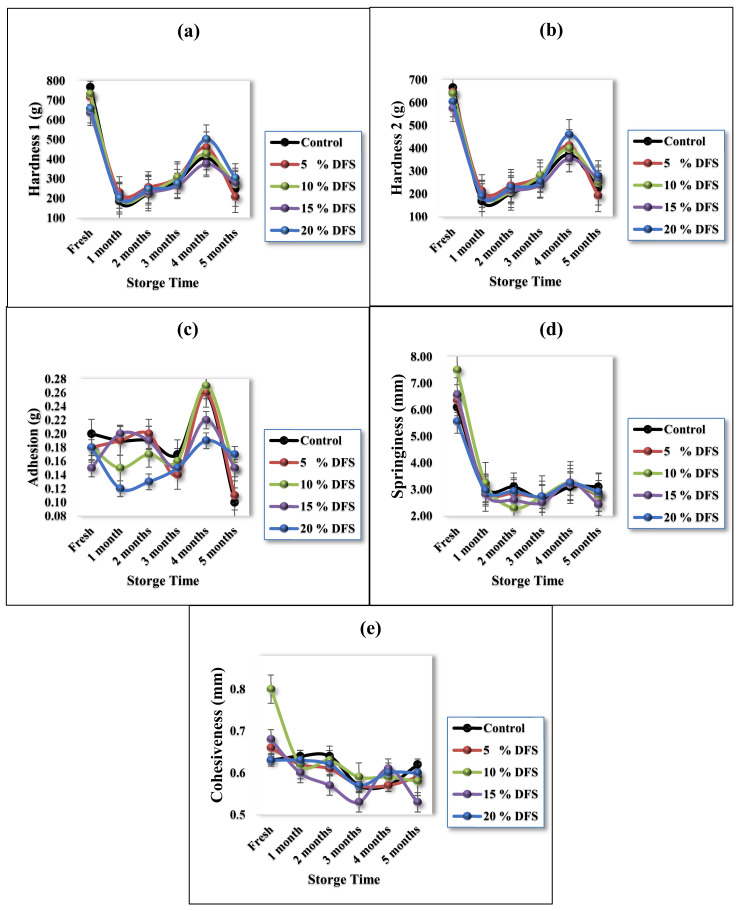
(**a**–**e**) Texture profile analysis of processed cheese with various ratios of DFS as a partial substitution for butter, in hardness 1, hardness 2, adhesion, springiness, and cohesiveness.

**Figure 5 foods-12-00679-f005:**
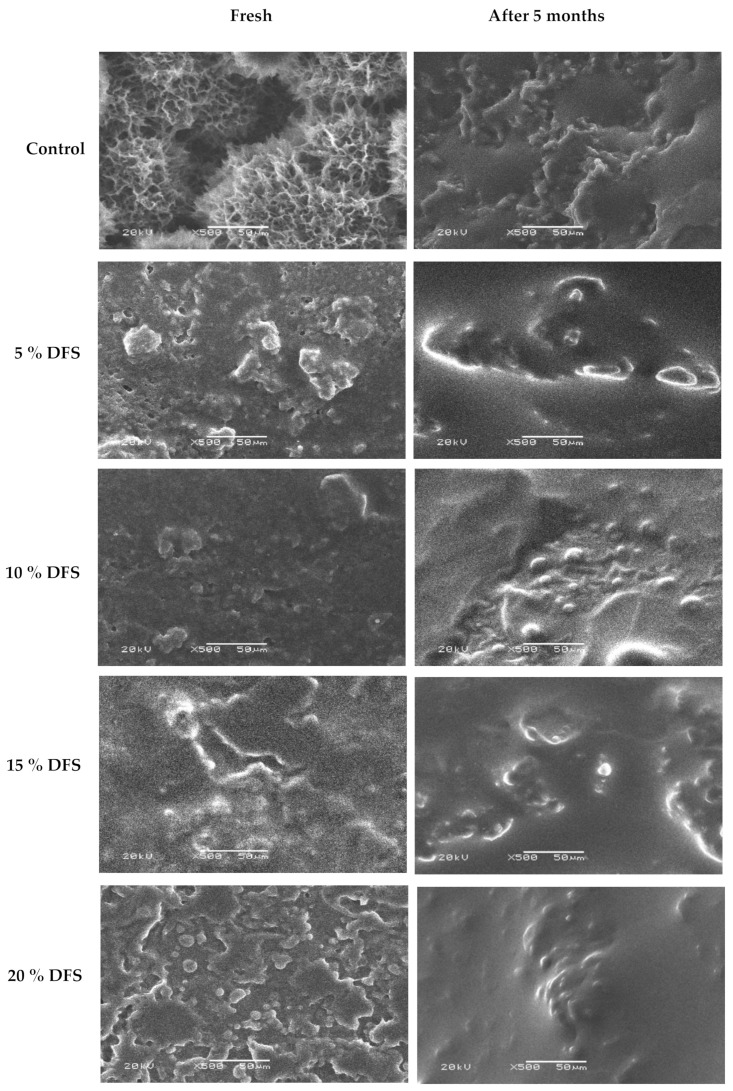
Scanning electron micrographs of the effect of DFS on the overall structure of block-type processed cheese when fresh and after five months of storage at room temperature. Magnitude 500×; 20 kV. Bar = 50 µm.

**Figure 6 foods-12-00679-f006:**
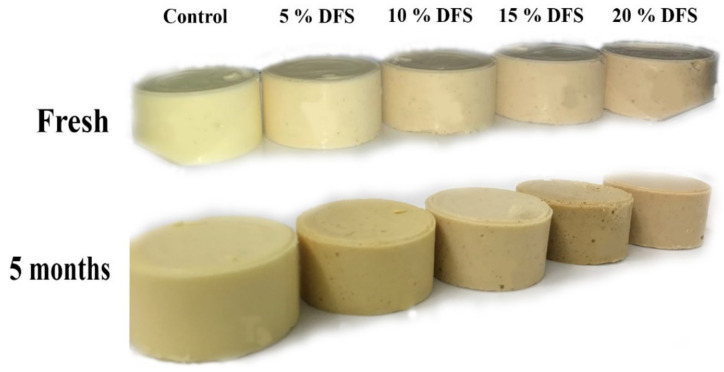
The visual appearance of block-type processed cheese fortified with DFS as a partial substitution for butter.

**Figure 7 foods-12-00679-f007:**
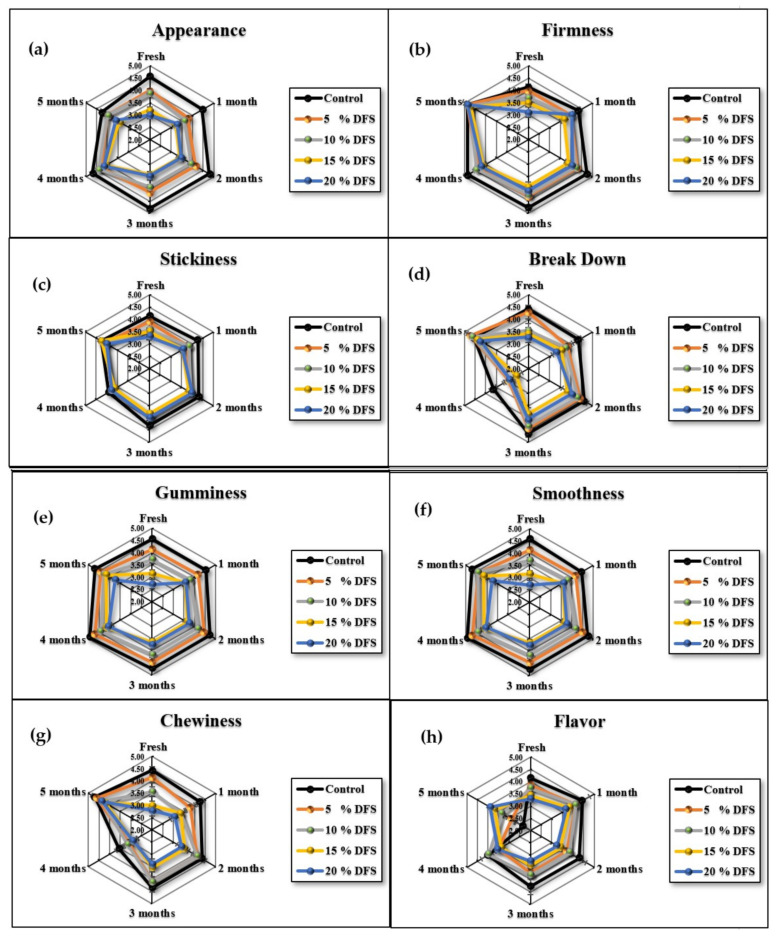
(**a**–**h**), Appearance, firmness, stickiness, breakdown, gumminess, smoothness, chewiness, and flavor of cheeses fortified with 5%, 10%, 15%, and 20% DFS as a partial substitution for butter during storage.

**Figure 8 foods-12-00679-f008:**
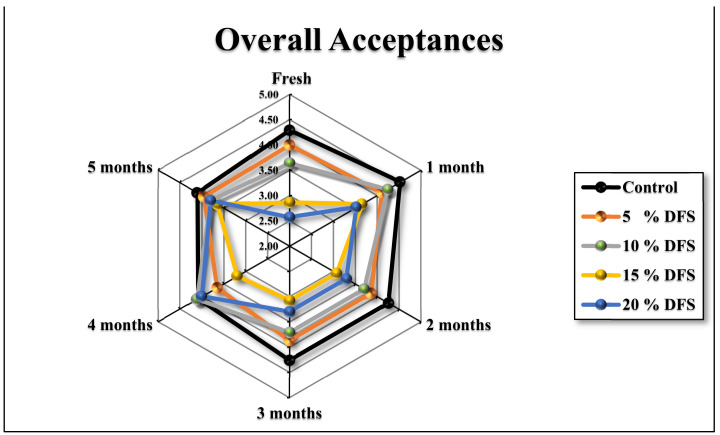
Overall acceptances of cheeses fortified with 5%, 10%, 15%, and 20% DFS as a partial substitution for butter during storage.

**Table 1 foods-12-00679-t001:** Chemical ingredients and experimental design of block-type processed cheeses.

Ingredients (g), (*w*/*w*)	Cheese Codes				
	Control	5% DFS *	10% DFS *	15% DFS *	20% DFS *
Butter	103.50	98.35	93.15	88.00	83.00
DFS	0.00	5.175	10.35	15.53	20.70
Standard cheddar cheese	540	540	540	540	540
Frozen cheddar cheese	300	300	300	300	300
Emulsifying salt (phosphate salts, Joha C)	27.30	27.30	27.30	27.30	27.30
Potassium sorbate	1.17	1.17	1.17	1.17	1.17
Nisin	0.117	0.117	0.117	0.117	0.117
Water	200	200	200	200	200
Total	1172.1	1172.1	1172.1	1172.1	1172.1

* The % of DFS was calculated as a partial replacement for butter compared with the amount of control butter.

**Table 2 foods-12-00679-t002:** Chemical and microbiological characterization of DFS.

Contents	Chemical Compositions g/100 g	Microbiological Analysis CFU/gm	
Moisture%	9.39 ± 0.10	Coliform	Nd
TS%	90.61 ± 3.01	Yeast and mold	<12
Protein%	4.86 ± 0.58 (5.36% DM)	*Escherichia coli*	Nd
Fat%	18.44 ± 1.72 (20.35% DM)	*Staphylococcus aureus*	Nd
Total Fiber%	65.95 ± 0.70 (71.99% DM)	*Clostridium perfringens*	Nd
Ash%	1.36 ± 0.01(1.50% DM)	*Bacillus cereus*	Nd

Nd: not detected; DM: dry matter.

**Table 3 foods-12-00679-t003:** Effect of different DFS% on color parameters during storage.

	Storage	Control	5% DFS **	10% DFS **	15% DFS **	20% DFS **
L* Value	Fresh	80.96 ± 0.66 ^Ba^	72.90 ± 0.85 ^Cb^	68.96 ± 2.54 ^Bcb^	65.86 ± 4.73 ^Bc^	64.45 ± 3.01 ^ABc^
	1 month	80.18 ± 1.14 ^Ba^	73.98 ± 1.41 ^BCb^	69.95 ± 1.50 ^Bc^	66.40 ± 2.47 ^Bd^	64.13 ± 1.72 ^Bd^
	2 months	79.44 ± 0.32 ^Ba^	73.11 ± 1.02 ^Cb^	70.22 ± 0.95 ^Bb^	64.92 ± 4.53 ^Bc^	64.43 ± 1.85 ^Bc^
	3 months	80.76 ± 1.34 ^Ba^	73.08 ± 0.84 ^Cb^	69.93 ± 0.73 ^Bc^	66.66 ± 1.99 ^Bd^	64.06 ± 1.18 ^Be^
	4 months	85.37 ± 0.10 ^Aa^	76.39 ± 0.99 ^Ab^	75.74 ± 2.15 ^Ab^	72.50 ± 0.66 ^Abc^	69.43 ± 4.21 ^Ac^
	5 months	80.16 ± 0.70 ^Ba^	74.00 ± 0.59 ^Bb^	68.36 ± 1.05 ^Bc^	63.76 ± 0.58 ^Bd^	63.04 ± 2.30 ^Bd^
a* Value	Fresh	−3.35 ± 0.04 ^Cd^	0.53 ± 0.10 ^Bc^	2.34 ± 0.32 ^Bb^	3.66 ± 0.82 ^Aa^	4.24 ± 0.23 ^Aa^
	1 month	−6.79 ± 0.16 ^Ee^	−2.99 ± 0.32 ^Dd^	−1.08 ± 0.28 ^Dc^	0.32 ± 0.46 ^Bb^	1.21 ± 0.28 ^Ca^
	2 months	−6.49 ± 0.30 ^Dd^	−2.70 ± 0.17 ^DCc^	−1.03 ± 0.17 ^Db^	0.85 ± 1.11 ^Ba^	1.28 ± 0.26 ^Ca^
	3 months	−6.62 ± 0.06 ^DEe^	−2.77 ± 0.11 ^DCd^	−0.90 ± 0.21 ^Dc^	0.49 ± 0.47 ^Bb^	1.40 ± 0.16 ^Ca^
	4 months	−7.39 ± 0.18 ^Fe^	−2.47 ± 0.29 ^Cd^	−0.29 ± 0.43 ^Cc^	1.47 ± 0.10 ^Bb^	2.64 ± 0.29 ^Ba^
	5 months	−2.74 ± 0.11 ^Be^	0.75 ± 0.23 ^Bd^	2.71 ± 0.06 ^Bc^	4.26 ± 0.03 ^Ab^	4.57 ± 0.18 ^Aa^
b* Value	Fresh	22.66 ± 1.00 ^Ca^	18.22 ± 0.43 ^Cb^	16.35 ± 0.20 ^Cc^	16.65 ± 0.52 ^Cc^	14.31 ± 0.35 ^Cc^
	1 month	25.20 ± 0.62 ^Ba^	20.94 ± 0.04 ^Bb^	18.60 ± 0.42 ^Bc^	17.66 ± 0.16 ^Bd^	16.70 ± 0.34 ^Be^
	2 months	25.39 ± 0.54 ^Ba^	20.84 ± 0.11 ^Bb^	18.72 ± 0.01 ^Bc^	17.52 ± 0.15 ^Bd^	16.51 ± 0.44 ^Be^
	3 months	24.87 ± 0.54 ^Ba^	20.69 ± 0.64 ^Bb^	18.74 ± 0.36 ^Bc^	17.77 ± 0.35 ^Bd^	16.53 ± 0.30 ^Be^
	4 months	23.40 ± 0.38 ^Aa^	22.74 ± 0.63 ^Ab^	22.26 ± 0.77 ^Ac^	20.65 ± 0.24 ^Ad^	19.56 ± 1.39 ^Ad^
	5 months	21.58 ± 0.15 ^Da^	17.99 ± 0.02 ^Cb^	16.46 ± 0.26 ^Cc^	15.43 ± 0.23 ^Cd^	14.22 ± 0.33 ^Ce^

The values are expressed as mean ± standard deviation (N = 3); significant differences between means in a row are indicated by different superscript uppercase letters (*p* < 0.05), and the means within a column indicated by superscript lowercase letters differ (*p* < 0.05). ** The % of DFS was calculated as a partial replacement for butter in comparison with the control amount of butter.

## Data Availability

Data is contained within the article.

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
