# Peer review of "Physicochemical and Sensory Properties and Shelf Life of Block-Type Processed Cheeses Fortified with Date Seeds (Phoenix dactylifera L.) as a Functional Food"

_foods, 2023, doi:10.3390/foods12030679_

Round 1
Reviewer 1 Report
The topic of the manuscript is interesting and performed analyses adequate . However, the manuscript is not well-written. There are many mistakes, e.g. moisture contents in date fruit powder is given as 9.39% and simultaneously total solids (TS) as 99.61%. Furthermore, the language of the manuscript needs intensive improvement as in the present form text is largely incomprehensible and unclear. Moreover, the information regarding the level of addition of date seed powder in the whole manuscript is misleading as data in Table 1 suggests that percentages were calculated as percentages of butter substitution (not per 100 g of cheese sample), but butter was only one of the cheese ingredient.
Author Response
We would like to thank the reviewer for the insightful reviews.
1- Results about TS % calculation was corrected from 99.61 ± 3.01 % to 90.61 ± 3.01 %. This was a typo in the original manuscript
2- English language has been thoroughly revised throughout the manuscript.
3- The DFS % of substitution changed as partial substitution of butter

Reviewer 2 Report
Comments and Suggestions of reviewer
The present manuscript “Physicochemical, Shelf-Life, and Sensory Properties of Block- 2 type processed cheeses fortified with Date seeds (Phoenix dac- 3 tylifera L.) as a Functional Food” treats with interest the effect of the date seeds incorporation of on the physicochemical, shelf-Life, microstructure, color and the Sensory properties of cheese.
We can observe the seriousness with which the authors treated the experiment. However, a number of mentions should be made:
Comment
In the article the approach used to evaluate the microbiological characteristics, chemical composition, and physical and functional potentials of DFS and to study the impact of these fortifications on physicochemical, texture, sensory characteristics, microstructure, and shelf life on the DFS Block-type processed cheeses
I think that authors should add the techno-functional properties of the final products such as water and oil binding capacity, foaming, emulsifying activities in order to confirm the quality of the novel food.
· Lines 85, then dried at “50 â—¦C” under vacuum, please rectify it.
· Lines 85, please specify the speed used.
· Line 120-121, 25 ± 120 2 0C, there is a mistake.
· Line 130, authors should (w/w) after « by replacement 5, 10, 130 15, and 20% ».
· What is “Emulsifying salt” in table 1?
· Please add the effect of each ingredient in the Block -type cheeses (Line 157).
· Line 2 in table 1 add (w/w) (line 136).
· In Figure 2 authors should add a reference to the flowchart or manufacture process.
· Line 208, authors should define the TS meaning.
· Table 2 “Chemical” in the title, please rectify it.
· In table 2 “Chemical compositions g /100 g”, please add relative to “dry matter”.
· In table 2 authors should standardize Nd or nd.
· Line 223-228, please check shape and spaces.
· Line 235, please add (w/w) after 5 %, 10 % 15 %, and 20 % DFS.
· Line 252, authors should check the spaces.
· Figure 3, authors should change the scale in order to improve the quality of the graphs.
· Line 292-293 and line 299, the units should be added.
· I think it's interesting to see the cross section in the microstructural properties of the cheese samples.
· In the conclusion authors mention that the final product, inspire clinical trials and in vivo including DFS. I think you this sentence should be modified as authors did not study the biological activity of the final product.

Author Response
Comments and Suggestions of reviewer
The present manuscript “Physicochemical, Shelf-Life, and Sensory Properties of Block- 2 type processed cheeses fortified with Date seeds (Phoenix dac- 3 tylifera L.) as a Functional Food” treats with interest the effect of the date seeds incorporation of on the physicochemical, shelf-Life, microstructure, color and the Sensory properties of cheese.
We can observe the seriousness with which the authors treated the experiment. However, a number of mentions should be made:
We would like to thank the reviewer for the insightful reviews.
Comment
In the article the approach used to evaluate the microbiological characteristics, chemical composition, and physical and functional potentials of DFS and to study the impact of these fortifications on physicochemical, texture, sensory characteristics, microstructure, and shelf life on the DFS Block-type processed cheeses
I think that authors should add the techno-functional properties of the final products such as water and oil binding capacity, foaming, emulsifying activities in order to confirm the quality of the novel food.
This investigation was designed as a pilot study. In future research will design all the properties, biological activity, and clinical study (added in conclusions)
- Lines 85, then dried at “50 â—¦C” under vacuum, please rectify it.
Added
- Lines 85, please specify the speed used.
The specification of the hummer mill manual is written high speed there is no specific speed written
- Line 120-121, 25 ± 120 2 0C, there is a mistake.
Corrected (25 ± 2 0C.)
- Line 130, authors should (w/w) after « by replacement 5, 10, 130 15, and 20% ».
Added
- What is “Emulsifying salt” in table 1?
Added (Phosphate salts, Joha C), also it’s written in the material section
- Please add the effect of each ingredient in the Block -type cheeses (Line 157).
Added
- Line 2 in table 1 add (w/w) (line 136).
Added
- In Figure 2 authors should add a reference to the flowchart or manufacture process.
Added
- Line 208, authors should define the TS meaning.
Added (total solids)
- Table 2 “Chemical” in the title, please rectify it.
Added
- In table 2 “Chemical compositions g /100 g”, please add relative to “dry matter”.
Relative to dry matter were added
- In table 2 authors should standardize Nd or nd.
Standardized
- Line 223-228, please check shape and spaces.
Modified
- Line 235, please add (w/w) after 5 %, 10 % 15 %, and 20 % DFS.
Added
- Line 252, authors should check the spaces.
Modified
- Figure 3, authors should change the scale in order to improve the quality of the graphs.
Changed
- Line 292-293 and line 299, the units should be added.
Added
- I think it's interesting to see the cross-section in the microstructural properties of the cheese samples.
Figure 5 represents the microstructural properties of the cheese samples instated of cross-section as the difference in the structure cant be visible. Due to the homogenization, cooking, emulsifier, and heating the cross-section was look the same inside as outside
- In the conclusion authors mention that the final product, inspire clinical trials and in vivo including DFS. I think you this sentence should be modified as authors did not study the biological activity of the final product.
Modified

Reviewer 3 Report
In the submitted manuscript the authors investigated the Physicochemical, Shelf-Life, and Sensory Properties of Block-type processed cheeses fortified with Date seeds (Phoenix dactylifera L.) as a Functional Food. The topic of the research was interesting, but the authors does not organize the manuscript well. The text of the article is confusing in some parts. The literature review of the article is very weak. The materials and methods are not well written. Some methods need a complete explanation. Some tests are needed to evaluate the shelf life of fortified processed cheese. The article's discussion is very weak in some parts. English and style require a careful reorganization. In my opinion, the overall output has several weak points. However, as a reviewer, I have a few comments:
Abstract:
- The abstract should be revised and restructured.
- The purpose of the study should be included in the abstract.
- The authors should present in the abstract the major findings of this study in order to make the article attractive and induce reader's interest to read the full text.
Keywords:
-It is recommended that Keywords should be different from the title.
Introduction:
- Line 43: "FAOSTAT"?? Please write completely.
- Line 43: "Phoenix dactylifera" in italic form.
- Line 51: Please remove figure 1 from the introduction section.
- In order to express the importance of the current study, it is necessary to explain what has already been done in this field. The authors should review the studies that have been conducted on the use of DFS in increasing the shelf life of foods.
Materials and methods:
- You should report all details of materials (supplier, purity and ….).
- Section 2.2: Please describe the preparation of samples.
- Please add the method of evaluating the microbiological characteristics of DFS in section 2.2.3.
- Table 1: Why are the amounts of DFS used in the cheese formula more than 5, 10, 15, and 20 %?
- Line 163: "profusely" ????
Section 2.6. Statistical analysis: I suggest describing which are the study factors and variables.
Results and discussion
- The quality of Figures 3 and 4 is very low.
- The authors should have reported the results of total viable count (TVC), yeast and mold count, as well as lactic acid producing bacteria (LAB) in DFS-enriched processed cheese during storage.
-If we look at Fig. 3(c), the acidity of all treatments increased during the storage period, which must be due to the increase in the amount of LAB. Why have these tests not been done?
- Fig. 3 (e): Why did the amount of fat in the control sample increase in the last month?
- Fig. 3: The results of pH are not consistent with the acidity in the treatments. The pH is almost the same in the first and last month, but the acidity in the first month is very different from the last month. Why?
Results and Discussion is very important part of each manuscript published. In presented manuscript, there is no discussion in some sections. Authors should discuss their results with other scientific papers.
- Section 3.7: Sensory evaluation is very important here. Therefore, the authors should show the results of each of the evaluated factors separately.
- The conclusions should be integrated with more detailed results summarizing all the study and must reflect the innovation of this study and the perspectives.
Author Response
In the submitted manuscript the authors investigated the Physicochemical, Shelf-Life, and Sensory Properties of Block-type processed cheeses fortified with Date seeds (Phoenix dactylifera L.) as a Functional Food. The topic of the research was interesting, but the authors does not organize the manuscript well. The text of the article is confusing in some parts. The literature review of the article is very weak. The materials and methods are not well written. Some methods need a complete explanation. Some tests are needed to evaluate the shelf life of fortified processed cheese. The article's discussion is very weak in some parts. English and style require a careful reorganization. In my opinion, the overall output has several weak points. However, as a reviewer, I have a few comments:
We would like to thank the reviewer for the insightful reviews.
Abstract:
- The abstract should be revised and restructured.
The abstract was revised and restructured
- The purpose of the study should be included in the abstract.
The purpose of the study added to the abstract
- The authors should present in the abstract the major findings of this study in order to make the article attractive and induce reader's interest to read the full text.
We added the most important findings because of the limitation of word count in the abstract as the journal instructions (200 words maximum)
Keywords:
-It is recommended that Keywords should be different from the title.
Added more keywords different from the title
Introduction:
- Line 43: "FAOSTAT"?? Please write completely.
Added (Statistics Division, Food and Agriculture Organization of the United Nations)
- Line 43: "Phoenix dactylifera" in italic form.
Modified to italic form
- Line 51: Please remove figure 1 from the introduction section.
Figure 1 moved from the introduction section to the end of the introduction
- In order to express the importance of the current study, it is necessary to explain what has already been done in this field. The authors should review the studies that have been conducted on the use of DFS in increasing the shelf life of foods.
The studies have been conducted on the date fruit seed were included in the introduction section and fig. 1 (references 7-15)
Materials and methods:
- You should report all details of materials (supplier, purity and ….).
Added
- Section 2.2: Please describe the preparation of samples.
The preparation of samples is presented in the flowchart as shown in Figure 2. While the preparation of DFS is presented in section 2.1.1
- Please add the method of evaluating the microbiological characteristics of DFS in section 2.2.3.
Added to section 2.2.3
- Table 1: Why are the amounts of DFS used in the cheese formula more than 5, 10, 15, and 20 %?
Because the % of DFS was calculated as a partial replacement for butter from the control butter amount (103.50 x 5 /100 = 5.175), (103.5 x 10 / 100 = 10.35), (103.5 x 15 / 100 = 15.53) and (103.5 x 20 / 100 = 20.70)
- Line 163: "profusely" ????
Corrected to previously This was a typo in the original manuscript
Section 2.6. Statistical analysis: I suggest describing which are the study factors and variables.
From the abstract section the desing of the expermantal are present the depended and independed variables
Results and discussion
- The quality of Figures 3 and 4 is very low.
Figures 3 and 4 were replaced from pic to orginal charts
- The authors should have reported the results of total viable count (TVC), yeast and mold count, as well as lactic acid producing bacteria (LAB) in DFS-enriched processed cheese during storage.
Due to heating the mixture to 95 C, there was no growth
-If we look at Fig. 3(c), the acidity of all treatments increased during the storage period, which must be due to the increase in the amount of LAB. Why have these tests not been done?
The increase in acidity is due to loss of moisture, not the increase of LAB Due to heating the mixture to 95 C, there was no growth
- Fig. 3 (e): Why did the amount of fat in the control sample increase in the last month?
The increase was not significant (P ≥0.05), and due to the loss of moisture. The same trend of the result of fat was published in ((Aly, E.S.; Saadany, K. el; Dakhakhny, E. el; Kheadr, E. Reduced-Fat and Substituted Block-Type Processed Cheeses; 2017; Vol. 13;.))
- Fig. 3: The results of pH are not consistent with the acidity in the treatments. The pH is almost the same in the first and last month, but the acidity in the first month is very different from the last month. Why?
The differences were not significant (P ≥0.05), and due to the loss of moisture
Results and Discussion is very important part of each manuscript published. In presented manuscript, there is no discussion in some sections. Authors should discuss their results with other scientific papers.
Since this is the first work to report the effects of DFS on the Physicochemical, Shelf-Life, and Sensory Properties of Block-type processed cheeses fortified with Date fruit seeds therefore, it is not possible to compare the results for these indexes with the literature because they are lacking.
- Section 3.7: Sensory evaluation is very important here. Therefore, the authors should show the results of each of the evaluated factors separately.
Added
- The conclusions should be integrated with more detailed results summarizing all the study and must reflect the innovation of this study and the perspectives.
Modified

Round 2
Reviewer 1 Report
Not all comments from a review were taken into account in the revised version, e.g. correction of TS% was done only in Table 2 but the sentence "The moisture, TS (total solids), protein, fat, fiber, and ash were 9.39 ± 0.10, 99.61 ± 3.01, 4.86 ± 0.58, 18.44 ± 1.72, 65.95 ± 0.70, and 1.36 ± 0.01, respectively" is not correct (moisture 9.39% and total solids 99.61%)?
The information regarding the level of addition of date seed powder in the whole manuscript is still confusing. The percentages were calculated as percentages of butter substitution (not per 100 g of cheese sample), but the information provided e.g. in the abstract "Different concentrations of DFS (0%, 5%, 10%, 15%, and 20%) were added to block-type processed cheese." suggests to the reader that the percentages are calculated per 100 g of cheese.
Author Response
We want to thank the reviewer for the insightful reviews.
1- The results of the TS % in the results and discussion section were corrected from 99.61 ± 3.01 % to 90.61 ± 3.01 %. This was a typo in the original manuscript
2- we added (as a partial substitution for butter) in the abstract and throughout all manuscript also in the tables and figures
Reviewer 2 Report
Comments and Suggestions of reviewer
The present manuscript “Physicochemical, Shelf-Life, and Sensory Properties of Block- 2 type processed cheeses fortified with Date seeds (Phoenix dac- 3 tylifera L.) as a Functional Food” treats with interest the effect of the date seeds incorporation of on the physicochemical, shelf-Life, microstructure, color and the sensory properties of cheese.
We can observe the seriousness with which the authors treated the experiment. I think that the manuscript has been improved but a number of mentions should be made.
· Figure 2:
In the manufacturing process of block-type processed cheeses, authors should add more details about the concentration of each compound, it means the concentration of all the additives and the ratios used compared to that of the cheese (w/w)
Specify the types of analyzes in the graphs (last box)
· Figure 3 (a-d) and figure 4 (a-e), authors should change the scale in order to improve the quality of the graphs:
Correct the y-axis
Draw the axes properly
Eliminate lines in the background of figures
It is necessary to improve the scale to visualize the spaced curves
Remove the title in the middle of the figures, just put the titles of the axes correctly with the units
Author Response
We would like to thank the reviewer for the insightful reviews
1 figure 2 added the %, and the type of analysis was added
2 figures 3 and 4 were modified according to the reviewer's recommendations
Reviewer 3 Report
-
Author Response
We would like to thank the reviewer for the insightful reviews